# Inulin from Globe Artichoke Roots: A Promising Ingredient for the Production of Functional Fresh Pasta

**DOI:** 10.3390/foods11193032

**Published:** 2022-09-30

**Authors:** Graziana Difonzo, Giuditta de Gennaro, Giusy Rita Caponio, Mirco Vacca, Giovanni dal Poggetto, Ignazio Allegretta, Barbara Immirzi, Antonella Pasqualone

**Affiliations:** 1Department of the Soil, Plant and Food Science (DiSSPA), University of Bari Aldo Moro, Via Amendola, 165/A, I-70126 Bari, Italy; 2Istituto per i Polimeri, Compositi e Biomateriali, Consiglio Nazionale delle Ricerche, Via Campi Flegrei, 34-80078 Pozzuoli, Italy

**Keywords:** waste reuse, inulin, high polymerization degree, functional pasta, glycemic index, prebiotics growth

## Abstract

Globe artichoke roots represent an alternative and sustainable source for inulin extraction and are well-noted for their technological and functional properties. Therefore, the aim of our study was to exploit inulin with high degree of polymerization as a replacement of durum wheat semolina for the production of functional fresh pasta. The effect of increased level of substitution (5, 10, 15%) on cooking, structural, sensory, and nutritional properties were evaluated and compared with a control sample consisting exclusively of durum wheat semolina. Inulin addition caused changes to internal structure as evaluated by scanning electron microscopy. The enriched samples showed a lower swelling index, an increasing cooking time, and values of cooking loss (2.37–3.62%), mainly due to the leaching of inulin into the cooking water. Cooked and raw enriched pasta was significantly darker and firmer than the control, but the sensory attributes were not negatively affected, especially at 5 and 10% of substitution levels. The increase of dietary fiber content in enriched pasta (3.44–12.41 g/100 g) resulted in a significant reduction of glycaemic index (pGI) and starch hydrolysis (HI). After gastrointestinal digestion, inulin-enriched pasta increased prebiotic growth able to significantly reduce *E. coli* cell density.

## 1. Introduction

Globe artichoke (*Cynara cardunculus* L. subsp. *scolymus* (L.)) is a perennial plant belonging to the Asteraceae family that is cultivated as a polyannual crop with vegetative propagation. Nevertheless, the length of the crop cycle, which negatively influences yields and the quality of the heads, has led artichoke growers to take an interest in the development of new seed-propagated cultivars for annual crops [1]. Italy represents one of the major producers, accounting for 33% of global production, followed by Spain, France, and Greece [2,3,4,5]. The edible part of the globe artichoke, commonly known as the “heart”, is highly appreciated worldwide and it is consumed fresh, canned, or frozen [6]. However, from the harvesting phase to the processing one, a large amount of waste and by-products (80–85% of the biomass) are produced: roots, stems, bracts, and leaves, which have shown to be a rich source of antioxidant compounds and dietary fiber, mainly as pectin and inulin [6,7,8].

Artichoke roots are an important agricultural waste that, at the end of the harvesting period, remains unexploited in the field [1]. At the same time, their richness in inulin, characterized by a high degree of polymerization [9], makes them an alternative and sustainable source, considering that chicory roots, Jerusalem artichoke, and dahlia are commonly used together to produce commercial inulin [10,11,12]. This fiber is the most abundant reserve polysaccharide after starch, and its structure is characterized by a mixture of oligo and/or polysaccharides consisting of a variable number of d-fructose units bonded with a β-(2 → 1) linkages with a terminal glucose residue. The degree of polymerization can range from 2 to 60 units, which directly influences its physicochemical and nutritional properties and its application as a food ingredient [9,12,13,14]. Generally, short-chain inulin is used as an alternative low-calorie sweetener, with its good solubility contributing to improving the mouthfeel, while long-chain inulin mixed with water or aqueous solution forms a particle gel network that can act as a fat replacer or texture modifier due to its lower solubility and good viscosity stability [12,15].

From the nutritional point of view, the peculiar bond configuration β-(2 → 1) confers to inulin a prebiotic character [16]. Inulin, indeed, reaches the colon unaltered, where it is fermented by beneficial bacteria such as *Lactobacillus* and *Bifidobacterium*, determining the release of short-chain fatty acids in the gut and a pH lowering which, in turn, enhances the absorption of minerals (Ca^2+^ and Mg^2+^), nutrients [4], and increases the functionality of colonocytes. Moreover, regular consumption of prebiotics has shown several health benefits like modulation of hyperglycemia, reduction of LDL cholesterol and serum lipids, prevention of colorectal cancer, and enhancement of immune system efficiency [17,18,19].

The consumer awareness about the relationship between health and food has led to the spread of functional products, i.e., those foods which contain bioactive compounds that, if taken in suitable quantities, have been shown to be able to prevent disease in addition to their nutritional functions [20,21]. In this regard, pasta seems to be particularly suitable for functional ingredients integration due to its easy preparation and its widespread consumption. Several studies have tried to improve the nutritional profile of pasta by adding soluble and insoluble fiber and evaluating their effect on quality properties [22,23,24,25,26,27,28,29,30].

However, the inclusion of dietary fiber could cause a deterioration of pasta quality in terms of cooking properties and sensory features due to the alteration of protein–starch network integrity, as observed by Foschia et al. [31]. Aravind et al. [32] found that the use of commercial inulin with a low degree of polymerization (DP) had a negative impact on firmness, cooking loss, and sensory acceptability of pasta; instead, the inulin with higher DP provided minimal impact. Similarly, Padalino et al. [33] added inulin at different DP and at two different concentrations (2 and 4%). In particular, the authors found interesting results in that the addition of inulin with a high DP, compared to the low one, had a greater disruptive effect on the starch–protein matrix. Peressini et al. [30] found an increase of firmness in pasta added with Barley Balance (BB), Psyllium seed husk (P) and BB–P, while a lower value than the control when inulin with high DP was used. Moreover, during the in vitro starch digestion, the pasta enriched with both inulin with high and low DP showed the highest reducing sugar release at 20 min compared to other fibers and control sample without fiber. Hence, Garbetta et al. [28] studied the effect of 4% addition of two different types of inulin, namely artichoke roots with high DP and chicory roots with low DP, observing good results in terms of sensory acceptability after the addition of inulin with high DP, suggesting the potential use of inulin-enriched spaghetti as a prebiotic food.

Therefore, an analysis of the literature revealed the influence of the DP of inulin on the technological and functional properties of foods, which needs to be further investigated based on the conflicting results in the literature.

In this framework, our study aimed to enhance the value of globe artichoke roots using them as an alternative source to extract inulin with a high degree of polymerization and their use as a functional ingredient for fresh pasta preparation. The effect of increasing the amount of the extracted inulin on structural, nutritional, and sensory properties of fresh pasta samples was evaluated.

## 2. Materials and Methods

### 2.1. Extraction and Characterzsation of Artichoke Root Inulin

#### 2.1.1. Extraction Process

Inulin extraction from artichoke roots (AR) was carried out according to Castellino et al. [10], with slight modifications. Artichoke roots powder (ARP) was mixed with water at a pH of 6.8 with a ratio solid to water of 1:16 (*w/w*). The extraction took place in a thermostated bath at 80 °C for 2 h, with periodic stirring every 15 min. Afterwards, the sample was filtered with a Buchner funnel using Whatman^TM^ (Darmstadt, Germany) filters with 11 µm of porosity, and then the filtrate was collected and submitted to precipitation phase through two cycles of freezing and thawing. The precipitate was centrifugated at 7500× *g* 15 min at 10 °C and the resulting pellet was washed by adding 10 mL of ethanol, centrifugated, dried overnight, and weighted.

Inulin yield was expressed in g per 100 g of ARP.
$$\mathrm{Yield}\;(\%)=\left(\;\frac{weight\;\left(g\right)\;of\;dried\;inulin}{weight\;\left(g\right)\;of\;ARP}\right)\times 100\;$$

#### 2.1.2. Moisture and Water Activity of AR Inulin Powder

Moisture was determined with a moisture analyzer (Mod. MAC 110/NP, Rodwang Wagi Elektroniczne, Radom, Poland) at 120 °C until constant weight. Water activity was measured using a hygrometer (Aqua Lab 100–240 V AC, Pullman, WA, USA). Each measurement was carried out in triplicate.

#### 2.1.3. Identification and Quantification of AR Inulin

Identification and quantification of AR inulin was conducted via high-performance liquid chromatography (HPLC) analysis using a 1260 infinity series chromatograph (Agilent Technologies, Santa Clara, CA, USA) equipped with a refractive index detector (RID) and a cationic exchange column 300 *×* 7.8 mm (Rezex^TM^ RCM-Monosaccharide Ca^2+^, 8 µm, Torrance, CA, USA). The analysis was conducted isocratically using Milli-Q water as a mobile phase with a flow of 0.6 mL/min, column temperature of 80 °C, and RID of 35 °C. For calibration, commercial inulin with a high degree of polymerization and purity ≥98.5% (FibrulineTM XL chicory root fibre, COSUCRA Groupe Warcoing SA, Warcoing, Belgium) was used as standard. Standard inulin solutions were filtered through a 0.45 µm nylon filter and injected in triplicate at different concentrations (0.25 mg/mL, 0.5 mg/mL, and 1.0 mg/mL). AR inulin was properly diluted, filtered, and injected. A calibration curve was obtained for concentration versus area, and AR inulin was identified by coincidences of retention time with standard inulin and quantified through the corresponding calibration curve.

#### 2.1.4. Polymerization Degree and Molecular Weight

Gel permeation chromatography was performed on extracted inulin. The analysis was carried out by using a GPC Max (Viscotek, Malvern, UK) system equipped with a TDA 305 detector (Refractive Index, Low Angle Light Scattering, Right Angle Light Scattering and Viscometer) and UV detector. We used a pre-column TSK PW_XL_ and TSK Gel GMPWXL (Tosoh Bioscience, Griesheim, Germany). The sample was dissolved at 3 mg/mL concentration and eluted in MilliQ water containing 0.2% NaN_3_. After complete dissolution, the sample was filtered on a membrane having 0.22 μm porosity. The injection volume was 100 μL, and the flow rate was 0.8 mL/min. The chosen method of analysis was universal calibration, using polyethyleneoxide (PEO) standards, ranging from 21.300 kDa to 420 Da. The measurements, performed at 35 °C according to the temperatures of columns and detectors, were ran for 60 min in duplicate.

The polymerization degree (DP) was calculated as follows [34]:M_n_ = 180 + 162 × (DP_n_ − 1)
and
M_w_ = 180 + 162 × (DP_w_ − 1)
where M_n_ and M_w_ are number average molecular weight and weight average molecular weight, respectively. 

### 2.2. Fresh Pasta Preparation and Characterization

#### 2.2.1. Experimental Design

Inulin-enriched fresh pasta was produced by substituting 5% (P5), 10% (P10), and 15% (P15) of durum wheat semolina (Mulino Martimucci, Altamura, Italy) with AR inulin powder. A control sample was produced with 100% durum wheat semolina. Durum wheat semolina and durum-wheat-semolina–inulin were mixed with an adequate amount of water and manually kneaded until obtaining a homogenous dough. The dough was left to rest for 30 min and then laminated (2 mm thickness) and cut with the iPasta (Imperia, Moncalieri, Italy) moulder machine to produce tagliatelle pasta. Afterwards, fresh tagliatelle were put on a wooden vessel, covered with a cotton cloth, and left to dry at room temperature until reaching a moisture content in a range of 26–28% and a_w_ values ranging from 0.92 and 0.97, according with Italian legal requirements for fresh pasta. Two different batches were produced.

#### 2.2.2. Cooking Properties

Pasta samples were cooked in boiling distilled water at 1:10 *(w/v)* pasta-to-water ratio, without the addition of salt, as reported by Pasqualone et al. [35]. The optimum cooking time (OCT) was determined by removing tagliatelle from boiling water every 30 sec, cutting them, and checking for the white and opaque core to disappear, according to the AACC 16–50 official method [36]. After cooking, pasta samples were drained and rinsed with distilled water and allowed to rest for 5 min. Cooking loss was evaluated by combining cooking and rinsing water, measuring total volume, putting 20 mL in tarred Petri dishes, and evaporating in an air-oven at 105 °C until reaching constant weight. The residue, scaled up to total volume, was expressed as a percentage of the original pasta sample weight. After cooking at their OCT, the samples were analyzed for their water absorption and swelling index (grams of water per gram of dry pasta) according with Bustos et al. [37]. Each determination was carried out in triplicate.
$$Water\;absorption\;\left(\%\right)=\frac{Wc-Wr}{Wr}\times 100\;\;\;\;\;\;\;Swelling\;index=\frac{Wc-Wd}{Wd}\;$$

W_r_ = weight of raw pasta (g)

W_c_ = weight of cooked pasta (g)

W_d_ = weight of dried pasta (g)

#### 2.2.3. Inulin Loss in Cooking Water

P5, P10, and P15 were cooked at their OCT with a pasta-to-water ratio of 1:10 (*w/v*). Afterwards, 15 mL of cooking water was collected and analyzed for its inulin content by a cationic exchange HPLC following the same method described in Section 2.1.3. The inulin cooking water content was expressed as grams of inulin loss during cooking at OCT of 100 g of fresh pasta. The analysis was carried out in triplicate.

#### 2.2.4. Color and Firmness Evaluation

Color of raw and cooked fresh pasta sample at its OCT (red index, corresponding to a*, yellow index, corresponding to b*, and luminosity index, corresponding to L*) was evaluated using a colorimeter CM-600d (Konica Minolta, Osaka, Japan) equipped with SpectraMagicX software. Six measurements were recorded for each pasta sample.

Firmness evaluation of raw and cooked pasta at its OCT was carried out using a Z1.0 TN (Zwick Roell, Ulm, Germany) equipped with a blade set with a guillotine and 50 N load cell. Pasta firmness was evaluated as the force required to cut 5 strands of pasta at a speed of 0.17 mm/s and was expressed as the maximum force (N) required to cut pasta strands. Eight measurements were recorded for each sample.

#### 2.2.5. Microstructure Determination

The microstructure and the surface characteristics of the pasta were studied with a Zeiss Sigma 300 VP (Carl Zeiss NTS GmbH, Oberkochen, Germany) field-emission gun scanning electron microscope (FEG-SEM) equipped with a secondary electrons detector (SE). The analyses were done under vacuum (<10^−4^ Pa), using an accelerating voltage of 20 kV, an aperture of 30 μA, and a working distance between 4 and 5 mm and magnification of 1000×. Fragments of the different pasta samples were glued onto an aluminium stub with carbon tape. Before the analysis, all the samples were carbon-coated in order to make the surface of the specimen conductive.

#### 2.2.6. Quantitative Descriptive Sensory Analysis

Sensory analyses were conducted by a panel of eight trained tasters (6 females and 2 males) to evaluate the sensory attributes according to Pasqualone et al. [35]. Pasta samples were coded by a three-digit number, cooked at their OCT, and evaluated for color, smell, taste, bulkiness, adhesiveness, hardness, and overall acceptability, using a structured scale ranging from 1 to 10. Color refers to the typical color of durum wheat pasta, where 1 = yellow color and 10 = brown; smell as perceived by olfaction and taste as perceived during mastication, which refer to the typical odor and taste of durum wheat pasta without anomalies, where 1 = very low and 10 = very high; bulkiness refers to adhesion of tagliatelle strands to each other, evaluated both visually and manually by pressing two tagliatelle together and determining the force required for detachment; stickiness, related to the organic matter released during cooking and still adhering to the surface of pasta, evaluated by pressing a single strand against the palate and determining the force require to remove it with the tongue; hardness, which is the resistance of cooked pasta to chewing measured while cutting the spaghetti strand using the front teeth. The analysis was carried out according to Pasqualone et al. [38], following the ethical guidelines of the Laboratory of Food Science and Technology of the Department of Plant, Soil, Food Science of University of Bari, Italy.

### 2.3. Functional Properties Determination

#### 2.3.1. Proximate Composition of Fresh Pasta

Protein (total nitrogen × 5.7), ash, and lipids content were determined using the AOAC method 979.09, 923.03, and 945.38 F, respectively [39]. Total dietary fiber content was determined by enzymatic gravimetric as described by the AOAC Official Method 991.43. Moisture content was determined by a moisture analyzer (Mod. MAC 110/NP, Radwag Wagi Elektroniczne, Radom, Poland) at 120 °C. The carbohydrate content was determined as difference. The determinations were carried out in triplicate.

#### 2.3.2. In Vitro Starch Hydrolysis

In vitro gastrointestinal digestion of fresh pasta and starch hydrolysis was determined according to Liljeberg et al. [40], simulating the in vivo digestion of starch. Briefly, aliquots of fresh pasta samples, cooked until the optimal cooking time and containing 1 g of starch (determined in cooked fresh pasta), were subjected to an enzymatic process (pancreatic amylase and pepsin-HCl), and the released glucose content was measured with D-fructose/D-glucose Assay Kit (Megazyme, Wicklow, Ireland). Simulated digests were dialyzed (cut-off of the membrane: 12,400 Da) for 180 min. Aliquots of dialysate, containing free glucose, and partially hydrolyzed starch were sampled every 30 min and further treated with amyloglucosidase. Then, free glucose was determined using the above-mentioned enzyme-based kit and finally converted into hydrolyzed (digested) starch in pasta. Control white wheat bread was used as the control to estimate the hydrolysis index (HI = 100). The predicted glycemic index (pGI) was calculated using the equation pGI = 0.549 × HI + 39.71 [41]. Each sample was analyzed in triplicate.

#### 2.3.3. Prebiotic Activity Assay

To evaluate the prebiotic activity of AR inulin-enriched fresh pasta, samples were previously subjected to in vitro gastrointestinal digestion according to the method used by Kamiloglu and Capanoglu [42], with slight modifications. The in vitro gastrointestinal digestion was performed, comprising of a pepsin-HCl digestion for 3 h at 37 °C (to simulate gastric digestion) and pancreatin digestion with pancreatin and bile salts for 3 h at 37 °C (to simulate small intestinal digestion). As reported by Caponio et al. [43], 10 mL of each fresh pasta extract was added to α-amylase (56 mg/mL) (Sigma-Aldrich Chemistry, St. Louis, MO, USA) and to 10 mL of pepsin solution composed of NaCl 125 mM/L + KCl 7 mM/L + NaHCO_3_ 45 mM/L + pepsin (Sigma-Aldrich Chemistry, St. Louis, MO, USA) at 3 g/L. Then, the pH was adjusted to 2 using HCl and incubated at 37 °C for 180 min in a water bath under shaking. After incubation, an aliquot of the gastric-digested extract was added in equal volume to an intestinal solution. The intestinal solution was simulated by dissolving 0.1 g/100 mL of pancreatin (Sigma-Aldrich Chemistry, St. Louis, MO, USA) and 0.15 g/100 mL bile salts (Oxoid™, Hampshire, UK). The pH was adjusted to 8 using NaOH and incubated at 37 °C for 180 min in a water bath under shaking. After incubation, an aliquot of intestinal-digested extract was ultra-filtrated with 3000 Da membrane (Vivaspin 20, Sartorius, Goettingen, Germany) to eliminate free carbohydrates. The retentate fractions were diluted in water and then filtered using 0.45 μm Whatman filter paper and further analyzed for prebiotics activities, as follows. 

Twenty-two probiotic strains of probiotics and one strain of *Escherichia coli* (*E. coli*) available in the culture collection of the Department of Plant, Soil, Food Science of University of Bari, Italy were used to carry out the experiments in fecal batches. The fecal medium (FM) was constituted as previously described [44] without the addition of glucose. This was labelled as FM (absence of carbohydrates), FMPC (FM + pasta not containing inulin), FMP5 (FM + pasta with 5% of inulin), FMP10 (FM + pasta with 10% of inulin), FMP15 (FM + pasta with 15% of inulin). For those fecal samples containing pasta, this was added in a ratio of 1:5 (*w/v*) in media after cooking and digestion was simulated as previously described. Viable probiotics and *E. coli* were inoculated in fecal media at a cell density of 7 UFC/mL (log_10_), measured through OD at 620 nm. Inoculated batches were incubated in anaerobic conditions for 36 h at 37 °C, under slight stirring (150 rpm). After the incubation, plate counts for lactic acid bacteria and *E. coli* were respectively made in De Man, Rogosa, and Sharpe agar (MRS) and Violet Red Bile Glucose agar (VRBGA). Both agar media were purchased from Oxoid Ltd. (Basingstoke, Hampshire, England, UK). Probiotic growth was also profiled in terms of ΔpH, as the difference between final (36 h) and initial (pH 7.0 ± 0.02) values of pH.

### 2.4. Statistical Analysis

The experimental data were subjected to one-way and two-way ANOVA, followed by a Tukey’s HSD test. The two-way ANOVA analysis was carried out considering the rate of substitution and the physical state of pasta (raw and cooked) as factors. Significant differences among the values of all the parameters were determined at *p* < 0.05 by the Minitab 17 Statistical Software (Minitab, Inc., State College, PA, USA, 2010).

## 3. Results and Discussion

### 3.1. Characteristic of AR Inulin Powder

Inulin extraction using hot water is considered the conventional extraction technique and the most-used [45]. Several factors can influence the extraction yield: temperature, time of extraction, solid/liquid ratio [46]. From the data collected in our study, the extraction yield, as the mean of ten measurements, was 23.37% ± 1.55, with a purity level of 89%, estimated in comparison to commercial inulin used as standard, which had a purity of 98.5%. The degree of polymerization (DP) is a fundamental parameter to characterize inulin, being directly correlated to its technological and nutritional properties. AR inulin has shown a DP_n_ and DP_w_ equal to 45 and 60, respectively, much higher than inulin extracted from Jerusalem artichoke (5–19), agave (5–13), and dahlia (17–23) [47,48,49]. This result highlights the possible use of AR inulin as a functional ingredient, since higher DP is associated with the improvement of technological properties in food products [10]. Among the factors that could affect the DP are plant, period of harvesting, storage period, and extraction process [50]. Moreover, the AR inulin powder has shown a moisture content equal to 6% and water activity of 0.40 ± 0.0, values which assure a high glass transition temperature, lower cohesiveness and, consequently, higher physical and microbiological stability [46,51].

Figure 1 shows the gel permeation chromatography (GPC) profile of AR inulin. Two different peaks were observed, the first peak in the range of 15–19 mL, relative to polysaccharides elution, and a second very sharp peak centred at 20.5 mL, presumably due to the elution of very short polysaccharide chains, monomers, and impurities present in the sample. The ratio between the two peaks was 75/25.

### 3.2. Quality Characteristics of Fresh Pasta

#### 3.2.1. Cooking Properties of Fresh Pasta

Pasta cooking properties are of great importanceforo ensuring acceptability by consumers. These properties were evaluated by considering parameters such as optimal cooking time (OCT), swelling index (SI), water absorption index (WAI), and solid cooking loss (CL) (Table 1). OCT was set for each pasta sample, observing a slight increase in pasta enriched with 10% and 15% of inulin. The same trend was observed by Foschia et al. [31] in pasta fortified by inulin with high DP and Simonato et al. [52] after the addition of *Moringa oleifera* L. leaves powder. However, the aforementioned results disagreed with other studies [32,33,53,54], where the addition of inulin caused a significant reduction of OCT due to the disruption of gluten network, which, in turn, caused an easier water penetration into starch granules. The discordant results could be attributed to the different production processes, as well as to the added fibrer and its intrinsic properties, which influence the interaction with other ingredients.

Regarding the WAI, no significant differences were found among the samples, whereas P10 and P15 showed a significant lower SI than control (PC). In accordance with our results, Naji-Tabasi et al. [55] and Attanzio et al. [56] noted a significant reduction of SI in pasta samples after the addition of wheat bran, mucilaginous seeds flour and *Opuntia* cladodes extract. Moreover, Renoldi et al. [57] stated that adding Psyllium fiber led to a lower peak value of storage modulus (G’) of dough, revealing differences in starch swelling of starch granules. The explanation for these results lies in the encapsulation of starch granules into fiber reticule, whose hydroxyl groups compete with starch and protein for water absorption, thus limiting the penetration of water into the starch granules and their consequent swelling [57,58,59]. On the contrary, the addition of whole barley flour to pasta formulation led to an increase of SI, which was related to the presence of β-glucans and their ability to absorb water [60]. Hence, different fibers could have a different effect on starch hydration and swelling.

CL refers to pasta resistance during cooking, and it is strictly connected with the strength of the protein network [61]. Good quality pasta should not have a CL higher than 7–8% [55]. In our study, CL ranged between 2.37% for the control sample (PC) and 3.62% for the P15. However, considering that inulin water solubility increases with high temperature [62], we also estimated the contribution of inulin to the observed CL. Specifically, inulin-imputable CL accounted for 1.01 ± 0.03 g/100 g of pasta in P5, 1.45 ± 0.07 g/100 g of pasta in P10, and 2.19 ± 0.12 g/100 g of pasta in P15. Consequently, making a difference between the CL values of P5, P10, and P15 and the corresponding inulin content found in cooking water, we could assert that in inulin-enriched samples, the solids different from inulin that leach into cooking water are lower than PC. Therefore, the probable formation of fibrous reticulum around starch granules [38] and additional hydrogen bonds and hydrophobic interactions between inulin and glutenin protein could provide extra support to the protein network, reducing the solid loss [63]. However, opposite results were found in several studies in the literature, in which the disruption of protein matrix due to fibers, mainly insoluble, promotes and allows the leaching of starch during cooking, causing an increase in cooking loss value [64,65,66].

#### 3.2.2. Color and Firmness of Fresh Pasta

Color, together with cooking and textural properties, is another key parameter for consumer acceptability: yellow color and high luminosity are associated with high-quality pasta [67]. Table 2 shows the color parameters in raw and cooked pasta samples. According to two-way ANOVA, both the cooking process and the addition of increasing percentages of AR inulin significantly affected the colorimetric parameters. Specifically, both raw and cooked pasta samples with higher rates of AR inulin substitution of durum wheat semolina caused a reduction of luminosity (L*) and yellow index (b*). Regarding cooking, the red index (a*) followed the same trend of L* and b*, whereas the opposite trend was observed for the increasing rate of substitution of durum wheat semolina with AR inulin. After the addition of insoluble fibers, Aravind et al. [68] found the same trend in terms of luminosity (L*), yellow, and red index, while Zarroug et al. [61] and Filipović et al. [21] reported an increase of L* values and a decrease of a* values for pasta enriched with commercial inulin, probably due to the added white color of the commercial inulin.

Firmness can be evaluated as the force necessary to cut the pasta strains, and it is strictly connected with the protein matrix development during pasta production and the hydration level of starch granules [64,67]. Cooked P10 and P15 showed a significantly higher firmness (7.37 ± 0.48 and 7.25 ± 0.37 N) than PC and P5 (5.85 ± 0.36 and 6.22 ± 0.26 N), which did not show any significant differences between them (Table 2). According to Chillo et al. [69] the mechanical properties of conventional and unconventional pasta are strictly connected with cooking properties. Therefore, the increase in firmness in cooked P10 and P15 can be reasonably linked to the lower values of the swelling index, supporting the hypothesis of pasta structure rearrangement as discussed in Section 3.2.1 [54]. Moreover, beta-glucan addition by Aravind et al. [70] and Peressini et al. [30] found a firmer pasta than control; however, significantly lower values of firmness were found by Peressini et al. [30] with the addition of 15% of inulin with a high degree of polymerization (DP = 23).

#### 3.2.3. Sensory Evaluation of Fresh Pasta

The results of the sensory evaluation are reported in Figure 2. The addition of larger amounts of AR inulin in fresh pasta formulations did not cause significant differences in sensory properties after cooking. The sensory scores for color and firmness have been consistent with instrumental evaluation, with all inulin-enriched pasta samples perceived as browner and P10 and P15 samples slightly firmer than PC. Although brightness and yellowness are linked to high-quality pasta, over the years, consumer attitude has been changing such that darker color is considered a positive trait, as it is associated with high-fiber products [25]. Significant differences were found in the taste of inulin-enriched samples than control, while only P10 odor resulted in a decreased perception rating compared to the typical odor of durum wheat pasta. In conclusion, P10 and P15 were statistically similar to P5; indeed, except for color and odor, no significant differences were highlighted for the other parameters considered. Therefore, it may be possible to add higher amounts of inulin without compromising the sensory properties of pasta.

#### 3.2.4. Microstructure

Figure 3 reports the SEM micrographs of cross sections of raw pasta (PC, P5, P10, and P15). Starch granules were well visible in all the samples, and the protein matrix did not appear disrupted. However, in P10 and P15, starch granules seem to be immerged in a more dense and compact structure. Moreover, P10 and P15 exhibit the presence of a reticulated structure, probably formed from the interaction of inulin with the protein network, which could have strengthened the pasta structure and thus giving a higher firmness to the cooked pasta as observed by the instrumental analysis.

### 3.3. Functional Properties of Fresh Pasta

#### 3.3.1. Proximate Composition of Fresh Pasta

Table 3 reported the proximate composition of fresh pasta. Significant differences were found among the samples for all the parameters considered. Specifically, inulin-enriched pasta showed a decline in protein content compared to the control (PC) due to a rise in total dietary fiber, which reached values of 3.44 g/100 g in P5, 8.16 g/100 g in P10, and 12.41 g/100 g in P15, exceeding the data observed by Padalino et al. [71] in pasta fortified with tomato byproducts. Therefore, the results in terms of total dietary fiber allow for labelling P5 as a “source of fibre”, while P10 and P15 could be labelled as a pasta having “high fibre content” [72], according to Reg. (EU) 1924/2006, enhancing the nutritional value of fresh pasta. Moreover, higher ash and lower lipid content were found in P10 and P15 than P5 and PC.

#### 3.3.2. In Vitro Starch Hydrolysis

In consideration the pivotal role of nutritional aspects for modern consumers and in recent scientific research aimed at producing foods with a lower glycemic index [73,74], cooked fresh pasta samples were analyzed for in vitro starch hydrolysis. As shown in Figure 4, the addition of AR inulin in fresh pasta promoted the decrease of HI and pGI. On average, the HI and pGI values of fresh pasta samples enriched with AR inulin (P5, P10, and P15) were statistically lower than the control (PC). Specifically, P5 reached a significantly (*p* < 0.05) lower value of pGI compared to the control (PC): 63.01 and 66.54, respectively. However, a higher concentration of AR inulin resulted in a further decrease of HI and pGI, as shown in P10 and P15. In fact, P10 has a statistically lower pGI value (58.42) compared to both the control (PC) and P5. Nevertheless, no significant differences were observed for P10 and P15 samples in terms of pGI and HI. Data demonstrate a lower pGI response following ingestion of AR inulin-enriched fresh pasta compared to the control.

Inulin, which is not digestible by humans, has interesting properties as a source of fermentable energy for some intestinal bacteria that produce short-chain fatty acids, which are essential for maintaining the intestinal homeostasis. In addition to stimulating digestion and regularity of intestinal transit, inulin may favor the presence of *Bifidobacterium* in the microbiota and, at the same time, decrease harmful bacteria [75,76]. Several studies have highlighted the beneficial role of inulin also in glycemic response [41,77,78,79]. Inulin—as a soluble fiber—helps keep blood sugar under control, since the fibers present in complex carbohydrates take longer to release the sugars present in the body: the slow release of glucose prevents glycemic peaks both upwards and downwards by balancing the energy intake and limiting the accumulation of fat due to the excess insulin [31,80]. The gelling effect of these fibers causes the formation of a film on the walls of the stomach and intestines with consequent lower absorption of fats and sugars [81]. In contrast to a previous study on chicory inulin-enriched pasta (from 2.5% to 10%) that did not reduce the pGI compared to the control [28], our results confirmed that AR was able to reduce the pGI and HI already at 5% of its concentration, due to the high DP of inulin used in our pasta samples.

#### 3.3.3. Evaluation of Effects Exerted by AR Inulin, Prebiotics, and Pathogen on Probiotic Growth

Herein, the prebiotic activity of inulin-enriched pasta was assessed in vitro in terms of probiotics growth. Furthermore, the inhibition of *E. coli* when co-cultured with probiotics was also determined. Compared to batches not containing carbohydrates (FM), the addition of pasta not containing inulin (FMPC) was sufficient to increase (~0.5 log_10_ CFU/mL) the cell density of all tested probiotics (Figure 5). This reflects the prebiotic contribution of fructans and arabinoxylans, which are non-digestible oligosaccharides naturally occurring in wheats [82], since their absence in gluten-free diets seems to affect the host microbiome and metabolome [83,84]. However, the presence of 3 g/L of inulin in FMP15 was able to further increase (>0.5 cycle) the cell density by 50% compared to the used probiotics (11 out of 22), while six strains (~27%) showed an increased cell density higher than one cycle. Besides the growth of probiotics, the acidification degree in batches followed the inulin concentration in a dependent manner. Values of ΔpH were, on average, 0.07 ± 0.06, 0.14 ± 0.11, and 0.20 ± 0.17 for FMP5, FMP10, and FMP15, respectively. No significant differences were found comparing the cell densities of *E. coli* in FMP to those of FMP5 (Figure 5). Oppositely, more than 50% (12 out of 22) of used probiotics were able to significantly decrease the cell density of *E. coli* in batches containing 3 g/L of inulin (FMP15). Meanwhile, ~36% of probiotics (eight strains) significantly decreased the *E. coli* cell density in batches containing 2 g/L of inulin (FMP10).

These results are in line with those previously stated by Kareem and co-workers [85], who reported that the combination of probiotics with prebiotics in vitro exhibited a great inhibition of pathogens due to a synergistic effect. Mechanisms based on microbial cross-feeding are largely distributed within the intestinal lumen [86]. A previous study concerning β-glucans-enriched pasta determined that 3 g of daily fiber supplementation was optimal to increase the saccharolytic metabolism in terms of SCFA profiling and improving the endothelial reactivity in healthy volunteers [87]. Therefore, although some used strains did not directly decrease *E. coli* growth, evidence suggests that additional taxa belonging to the human gut microbiota (e.g., clostridial cluster IV, XIVa, and *Bifidobacterium*) can support the metabolism of inulin in SCFA in vivo, eliciting a boosted effect that contributes to the host’s intestinal homeostasis [88].

## 4. Conclusions

The obtained results show that inulin from globe artichoke represents a promising functional ingredient in terms of technological and nutritional properties. As a matter of fact, adding inulin determined changes in the structure of raw and cooked pasta, which results more compact and firmer than control. The addition of inulin resulted in increasing the optimal cooking time, reducing the swelling index and increasing cooking loss (the latter due to inulin leaching in the cooking water). In terms of color, inulin-enriched samples (P5, P10, and P15) showed lower L* and b* and higher a* than control. The sensory properties did not substantially change compared to control. From the nutritional perspective, the ability of inulin to slow down the release of glucose in the blood was assessed, showing that a significant reduction of HI and pGI occurred in inulin-enriched pasta compared to control. Moreover, the in vitro prebiotic assay demonstrated that FMP10 and FMP15, containing 2 g/L and 3 g/L of inulin, respectively, significantly increased the cell density of prebiotics able to inhibit the *E. coli* growth. Therefore, the promising results obtained highlight the possibility to upcycle, in a circular perspective, the artichoke roots from agricultural waste to a valuable food ingredient that is useful for preparing high-added-value food products that could satisfy the increasing demand of consumers for more sustainable and functional foods.

## Figures and Tables

**Figure 1 foods-11-03032-f001:**
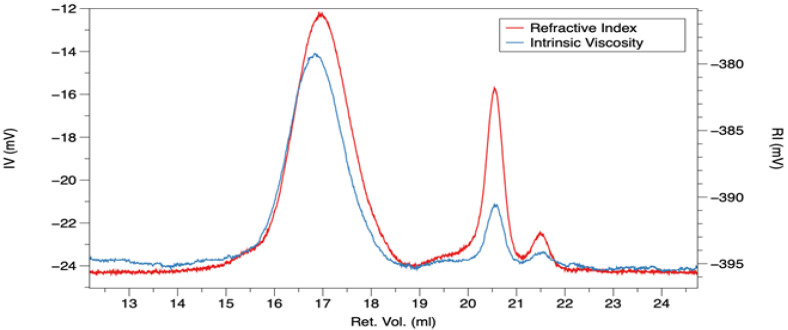
Gel permeation chromatography (GPC) profile of artichoke roots inulin, considering the refractive index (red line) and the intrinsic viscosity (blue line).

**Figure 2 foods-11-03032-f002:**
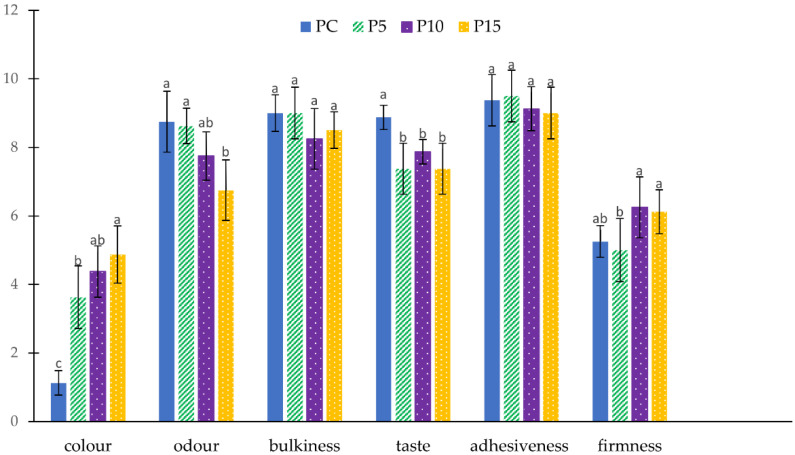
Sensory characteristics of control (PC) and inulin-enriched (P5, P10, P15) fresh pasta. Values are expressed as mean ± standard deviation; different letters for each parameters mean significant statistical differences (*p* < 0.05) to one-way ANOVA followed by Tukey’s HSD test.

**Figure 3 foods-11-03032-f003:**
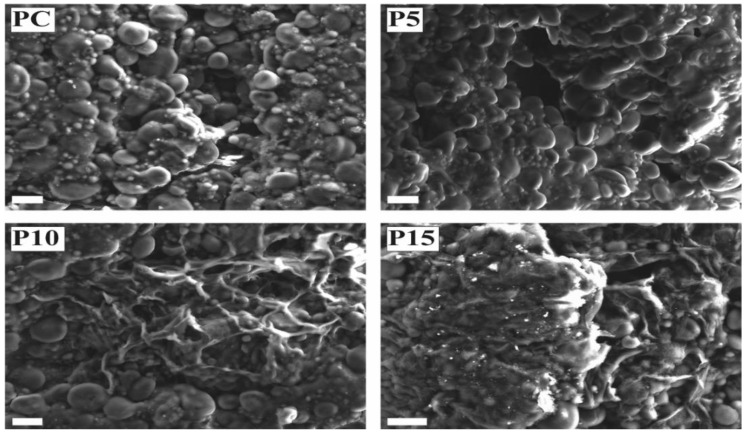
SE-SEM micrographs of control (PC) and inulin enriched (P5, P10, P15) fresh pasta (white bar = 20 µm).

**Figure 4 foods-11-03032-f004:**
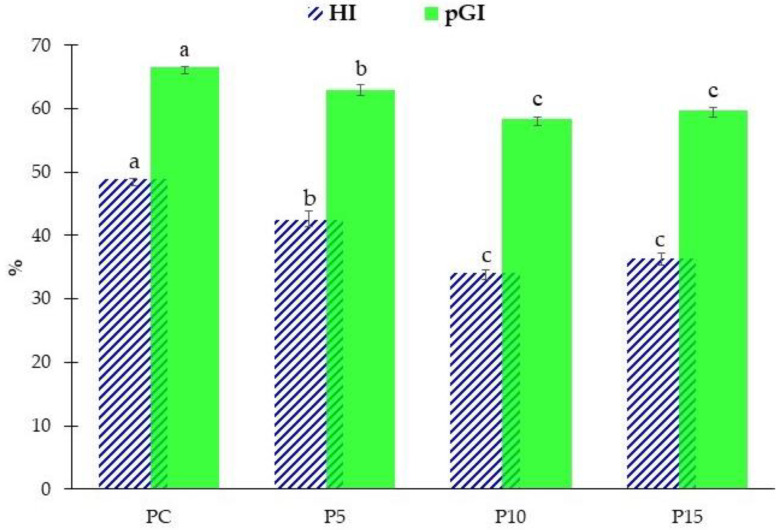
Results of hydrolysis index (HI) and predicted glycemic index (pGI) of control (PC) and inulin-enriched (P5, P10, P15) fresh pasta. The values represent means of triplicates ± standard deviation; different letters indicate significant differences (*p* < 0.05) according to one-way ANOVA followed by Tukey’s HSD test.

**Figure 5 foods-11-03032-f005:**
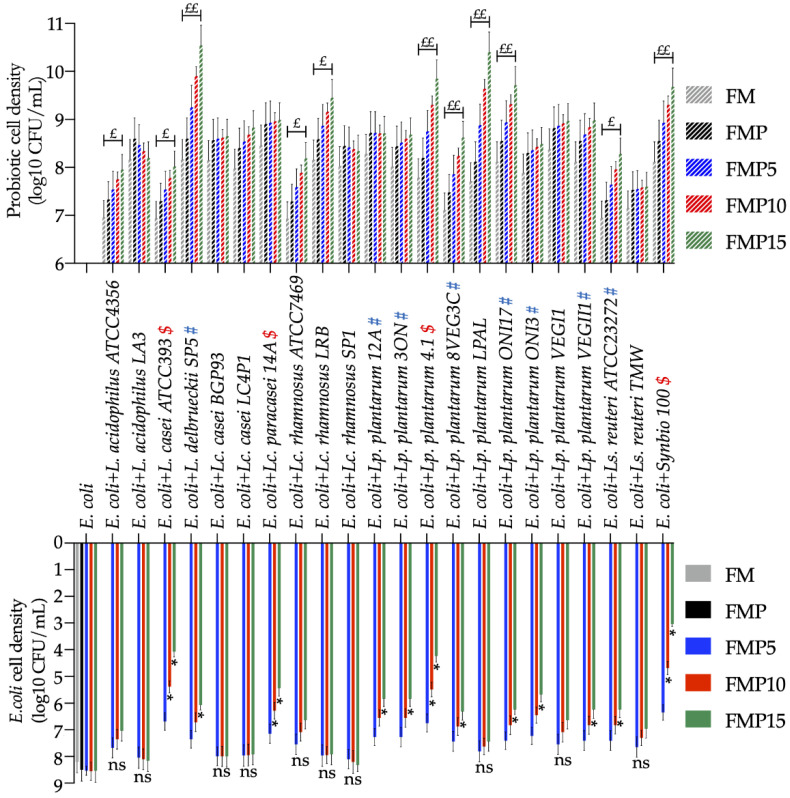
In vitro prebiotic assay showing the cell density of 22 probiotic lactic acid bacteria co-cultured with *E. coli* in fecal batches not containing carbohydrates (FM) or made with the addition of pasta without inulin (FMPC) and inulin-enriched pasta at 5 (FMP5), 10 (FMP10), and 15% (FMP15). “£” means >0.5 log10 cycle increased cell density (CFU/mL) of probiotic lactic acid bacte-ria in FMP15 versus FMP; “££” means >1 log10 cycle increased cell density (CFU/mL) of probiotic lactic acid bacteria in FMP15 versus FMP. “#” and “$” indicate those probiotics that had respective-ly determined a significant decrease of *E. coli* cell density in FMP15 or both FMP15 and FMP10. “*”: significant decrease of *E. coli* cell density compared to FMP; “ns”: no significant differences.

**Table 1 foods-11-03032-t001:** Cooking properties of fresh pasta.

Sample	OCT (min)	WAI (g/100 g)	SI	CL (g/100 g)	IL (g/100 g)
PC	6.30	73.20 ± 0.01 a	1.56 ± 0.03 a	2.37 ± 0.05 d	-
P5	6.30	73.24 ± 0.10 a	1.53 ± 0.04 ab	2.70 ± 0.14 c	1.01 ± 0.03 c
P10	6.45	72.51 ± 0.62 a	1.48 ± 0.03 b	3.11 ± 0.07 b	1.45 ± 0.07 b
P15	6.45	73.24 ± 0.20 a	1.47 ± 0.04 b	3.62 ± 0.06 a	2.19 ± 0.12 a

PC, control pasta without inulin addition; P5, pasta with 5% of inulin added; P10, pasta with 10% of inulin added; P15, pasta with 15% of inulin added. OCT, optimal cooking time; WAI, water absorption index, SI, swelling index; CL, cooking losses; IL, inulin losses. The values represent means of triplicates ± standard deviation; different letters in the same column mean significant statistical differences (*p* < 0.05) to one-way ANOVA followed by Tukey’s HSD test.

**Table 2 foods-11-03032-t002:** Color and firmness of fresh pasta.

		L*	a*	b*	Firmness
Raw pasta	PC	79.65 ± 0.09 a	2.21 ± 0.07 e	33.57 ± 0.46 a	18.25 ± 0.45 b
	P5	74.57 ± 0.19 c	3.21 ± 0.02 c	30.49 ± 0.26 b	19.63 ± 0.50 a
	P10	72.02 ± 0.38 e	3.89 ± 0.10 b	28.52 ± 0.19 c	19.89 ± 0.05 a
	P15	70.72 ± 0.25 f	4.48 ± 0.08 a	27.62 ± 0.09 cd	19.77 ± 0.23 a
Cooked pasta	PC	78.89 ± 0.43 b	0.30 ± 0.02 f	27.11 ± 0.56 d	5.85 ± 0.36 d
	P5	73.69 ± 0.18 d	2.00 ± 0.11 e	24.13 ± 0.41 e	6.22 ± 0.26 d
	P10	72.24 ± 0.06 e	2.82 ± 0.11 d	24.18 ± 0.33 e	7.37 ± 0.48 c
	P15	68.12 ± 012 g	3.79 ± 0.10 b	22.29 ± 0.43 f	7.25 ± 0.37 c
*p*-value	P *C	<0.0001	<0.0001	<0.0001	<0.05

PC, control pasta without inulin addition; P5, pasta with 5% inulin added; P10, pasta with 10% inulin added; P15, pasta with 15%= inulin added. P, percentage of inulin addition; C, cooking process. Values are expressed as mean of ± standard deviation; different letters in the same column mean significant statistical differences (*p* < 0.05) according to two-way ANOVA.

**Table 3 foods-11-03032-t003:** Proximate composition of fresh pasta (g/100 g).

Parameters	PC	P5	P10	P15
Ash	0.41 ± 0.02 b	0.41 ± 0.01 b	0.49 ± 0.01 a	0.50 ± 0.01 a
Protein	10.75 ± 0.03 a	9.71 ± 0.44 b	9.25 ± 0.02 b	9.27 ± 0.08 b
Total dietary fiber	1.47 ± 0.04 d	3.44 ± 0.10 c	8.16 ± 0.12 b	12.69 ± 0.08 a
Lipid	0.23 ± 0.02 a	0.10 ± 0.01 b	0.06 ± 0.01 c	0.08 ± 0.01 c
Moisture	28.21 ± 0.17 a	27.00 ± 0.21 b	26.98 ± 0.10 b	28.24 ± 0.30 a
Carbohydrates	58.94 ± 0.22 a	59.34 ± 0.25 a	55.07 ± 0.02 b	49.20 ± 0.42 c

PC, control pasta without inulin addition; P5, pasta with 5% of inulin added; P10, pasta with 10% of inulin added; P15, pasta with 15% of inulin added. The values represent means of triplicates ± standard deviation; different letters indicate significant differences; different letters in the same column mean significant statistical differences (*p* < 0.05) to one way ANOVA followed by Tukey’s HSD test.

## Data Availability

The data will be available on request.

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
