# Peer review of "Inulin from Globe Artichoke Roots: A Promising Ingredient for the Production of Functional Fresh Pasta"

_foods, 2022, doi:10.3390/foods11193032_

Round 1

Reviewer 1 Report

The manuscript is about the application of inulin to improve the quality of fresh pasta in terms of cooking, structural, sensory and nutritional characteristics. The manuscript is well-written by the authors, the abstract and introduction provide sufficient background information, the research design is appropriate related to the topic, the methodology is well-described, and the results are adequately presented and discussed.

However, I would like to suggest as follows: 

1. English language and style are fine/minor spell checks are required. Be consistent in using UK English or US English, choose either one.

2. There are minor corrections in terms of grammar and sentence structure. Kindly please refer to the attached file.

3. Re-check all references cited, they should be consistent with journal format/style.

Therefore, it is suggested to give a minor revision. Please refer to the attached file for further detailed comments that need improvement. Thank you.

Author Response

Dear reviewer,

find attached our revisions.

Bests

GD

Reviewer 2 Report

This study investigated the effects of inulin from globe artichoke roots on the physicochemical properties of functional fresh pasta. The manuscript is well written, and the results are clear and interesting.

Comments:

1. The authors should add the "Materials" Section.

2. Line 145; Mw/Mn, not Mw/Mn.

3. Line 154; “tagliatelle” does not need to be in italics (please correct the entire manuscript).

4. Line 163; Please add the reference.

5. Line 172-175; Wr, Wc, and Wd, not Wr, Wc, and Wd.

6. Line 193; Please write the magnification.

7. Line 195; Sigma

8. Line 397 and 398; The odour of P15 was significantly lower that other treatments.

Author Response

(The authors gave the same response as above.)

Reviewer 3 Report

The paper concerns the characterization of fresh durum wheat pasta enriched with inulin extracted from the globe artichoke roots. The article is well written and conceived and deals with different aspects starting from the extraction of inulin to the technological and sensory characterization of cooked pasta and the effect of inulina on the microbioma composition  and fits well into the highly topical topic of circular economy.

Nevertheless minor revisions and explanations are necessary:

L.29: What do you mean by 'perennial'? That it re- grows every year without replanting it? If so, why do you have to remove the roots from the soil after harvest?

L.205: What do you mean by 'structured' scale?

L.215: which are the ethical guidelines?

L.220: Why did you use the conversion factor 6.25 instead of 5.7, which is the standard use for wheat? Indeed, in Table 3 the protein content of Pasta Control is unusually high.

L. 315-318: Please add some hints of discussion to possible interpretate this discordant behaviours. Could you hypothesize a difference between dry and fresh pasta?

L. 328: did you mean barley?

L.330: Could you hypothesize a difference between soluble and insoluble fiber?

L.332: What did you intend by 'pasta resistance'?

L. 334-347: Authors should try to give possible explanations for these' opposite results'

Table1: Please insert units of measurements of the different parameters analyzed.

Table 1: Please insert the number of replications in the legend

Table 1: Why dont't you have SD for the OCT?

Table 3: Please insert units of measurement

Table 3: Why did you choose to present results about proximate composition of fresh pasta and not of the cooked one?

L. 460: might this be due to the DP of your extract?

L. 513: Did you mean P10 and P15?

Author Response

(The authors gave the same response as above.)
